# Effect of Betaine and Arginine on Interaction of αB-Crystallin with Glycogen Phosphorylase *b*

**DOI:** 10.3390/ijms23073816

**Published:** 2022-03-30

**Authors:** Tatiana B. Eronina, Valeriya V. Mikhaylova, Natalia A. Chebotareva, Kristina V. Tugaeva, Boris I. Kurganov

**Affiliations:** Bach Institute of Biochemistry, Federal Research Centre “Fundamentals of Biotechnology” of the Russian Academy of Sciences, Leninsky pr. 33, 119071 Moscow, Russia; mikhaylova.inbi@inbox.ru (V.V.M.); n.a.chebotareva@gmail.com (N.A.C.); kri94_08@mail.ru (K.V.T.); kurganov@inbi.ras.ru (B.I.K.)

**Keywords:** HspB5, glycogen phosphorylase *b*, chemical chaperones, aggregation, oligomeric structure

## Abstract

Protein–protein interactions (PPIs) play an important role in many biological processes in a living cell. Among them chaperone–client interactions are the most important. In this work PPIs of αB-crystallin and glycogen phosphorylase *b* (Ph*b*) in the presence of betaine (Bet) and arginine (Arg) at 48 °C and ionic strength of 0.15 M were studied using methods of dynamic light scattering, differential scanning calorimetry, and analytical ultracentrifugation. It was shown that Bet enhanced, while Arg reduced both the stability of αB-crystallin and its adsorption capacity (AC_0_) to the target protein at the stage of aggregate growth. Thus, the anti-aggregation activity of αB-crystallin increased in the presence of Bet and decreased under the influence of Arg, which resulted in inhibition or acceleration of Ph*b* aggregation, respectively. Our data show that chemical chaperones can influence the tertiary and quaternary structure of both the target protein and the protein chaperone. The presence of the substrate protein also affects the quaternary structure of αB-crystallin, causing its disassembly. This is inextricably linked to the anti-aggregation activity of αB-crystallin, which in turn affects its PPI with the target protein. Thus, our studies contribute to understanding the mechanism of interaction between chaperones and proteins.

## 1. Introduction

Protein–protein interactions (PPIs) play an important role in many biological processes in any living cell, where they are subjected to various influences such as subcellular localization, competitive interaction with other cellular factors, oligomerization/association, and post-translational modification [1]. The function and activity of the protein in most cases changes depending on its oligomeric state and its interactions with other proteins. PPIs differ depending on the composition, affinity, and whether the association is permanent or transient. PPIs can be permanent for substances that have a high affinity towards each other, such as an enzyme–inhibitor and an antibody–antigen. In contrast, transient interactions require the ability to change the affinity between proteins. Since a change in quaternary structure of an enzyme is often associated with biological function or activity, transient PPIs are important biological regulators that have a significant effect on the kinetic parameters of the enzyme [2].

Compared with other PPIs, molecular chaperone–target protein interactions are particularly weak and transient [3]. The relatively weak affinity and poor shape complementarity of chaperone–client interactions seem to allow the chaperone to recognize a wide range of different sequences [4,5,6] and adapt to a changing and adaptable proteome [7].

The molecular chaperones perform important biological functions of modulating protein homeostasis under constantly changing environmental conditions through PPIs with their client proteins [8] using both electrostatic and hydrophobic surfaces [9]. The molecular chaperones are central mediators of protein homeostasis. In this role, they engage in widespread PPIs with each other and with their client proteins. Together these PPIs form the backbone of a network that provides proper monitoring of protein folding, transport, quality control, and degradation [3].

Small heat shock proteins (sHsps) belong to the family of molecular chaperones and protect cells against various types of stress, especially heat stress, recover damaged proteins in the cell [10], and play an important role in maintaining cellular proteostasis [11,12]. One of the most important functions of sHsps is to prevent protein aggregation by binding of non-native or misfolded protein molecules and holding them in a folding-competent conformation. In addition, sHsps are involved in the regulation of many cellular processes and help maintain protein homeostasis [13]. There is growing evidence that the PPIs among sHsps can be reconfigured in disease settings. It is perhaps not surprising that sHsps and their functions are associated with a plethora of diseases including neurodegenerative diseases, multiple sclerosis, and cancers [14,15,16]. Therefore, the search for small molecule modulators that affect not only the chaperones themselves but also PPIs among chaperones and their clients is getting more and more attention of pharmacology and chemical biology.

αB-crystallin (αB-Cr or HspB5) is one of the small heat shock proteins. It is widespread in all tissues, but its concentration is especially high in the eye lens, where it interacts with αA-crystallin (HspB4) and forms a native complex, α-crystallin [17]. αB-Cr has a dynamic quaternary structure that allows it to form polydisperse assemblies of subunit-exchanging oligomers with chaperone-like activity [18]. The activity of the chaperone is regulated by a shift in equilibrium between oligomeric forms. An increase in chaperone activity is associated with the formation of small-sized oligomers (monomer and dimer) [18,19,20]. The equilibrium between different oligomeric forms of αB-Cr and chaperone effectiveness is very sensitive to many factors, such as the rate of the target protein aggregation, nature of the aggregation, temperature, the presence of ions, chemical chaperones, etc. [19,21,22,23]. Dysregulation of αB-Cr function underlies many human diseases [22].

In vivo protein chaperones work together with chemical ones, influencing each other’s activity. Chemical chaperones can affect not only PPIs between sHsp and client protein but also can affect each protein individually. They can directly control the stability of proteins, indirectly regulate protein homeostasis in the cell by affecting the activity of molecular chaperones, and promote the action of protein chaperones during protein refolding, which cannot occur in the presence of only one molecular chaperone [24]. Physiological concentrations of betaine (Bet) in *E coli* (up to 1 M) activate protein chaperones complexes (GroEL + GroES) and Hsp100 in correct refolding of urea-unfolded malate dehydrogenase [25]. A possible mechanism for such activation may include stabilization of the final product or specific activation of protein chaperones [26].

Arginine (Arg) regulates the anti-aggregation activity of some sHsps, such as α-crystallin [27,28], HspB5 [29] or HspB6 [30], increasing hydrophobic surfaces that results in an increase in the anti-aggregation activity of sHsps. However, sometimes as in case of aggregation of bovine catalase at 55 °C, 0.1 M Arg reduces activity of αB-Cr [29]. Moreover, sometimes Arg acts on PPIs in two opposite ways depending on the environmental conditions [31]. Thus, the effect of Arg on the chaperone activity of αB-Cr is the target protein-specific.

The concentration and balance between various additives may directly modulate the chaperone activity of protein chaperones and affect the PPIs between the chaperone and the client protein in cells under stress [25]. Therefore, understanding the mechanisms of the effect of chemical chaperones on the activity of protein ones and their PPIs is an urgent task. Moreover, little is known about the stoichiometry of mammalian sHsp–target protein complexes [32].

In this work, we studied the effect of Arg and Bet on αB-Cr and its PPIs with target protein and tried to determine the initial stoichiometry of the chaperone–client complex and the effect of Arg and Bet on this value at different stages of the enzyme aggregation. For the research, we chose a well-studied test system based on the thermal aggregation of glycogen phosphorylase *b* (Ph*b*) at 48 °C [33,34]. Previously it was shown that Arg and Bet acted in opposite directions on Ph*b* aggregation. Arg increased Ph*b* aggregation [35], while Bet [36] protected enzyme from aggregation. Therefore, it was interesting to study how these two chemical chaperones would act on PPIs between αB-Cr and target protein.

## 2. Results

### 2.1. Effect of αB-Crystallin on Phb Aggregation in the Presence of Chemical Chaperones

Figure 1A shows the kinetic curves of Ph*b* aggregation at 48 °C and ionic strength (*IS*) of 0.15 M (0.3 mg/mL; 0.03 M Hepes, pH 6.8, containing 0.15 M NaCl, 0.5 mM DTT) obtained by dynamic light scattering (DLS) in the absence of additives, in the presence of 0.05 mg/mL αB-Cr, and in the presence of 200 mM Bet or 100 mM Arg in a mixture with 0.05 mg/mL αB-Cr. It was shown that αB-Cr inhibits the aggregation of the model protein (Figure 1A, black and red curves), significantly reducing the light scattering intensity (*I* − *I*_0_) of the protein solution. However, the addition of chemical chaperones (Figure 1A, green and blue curves) results in an earlier increase in *I* − *I*_0_ values, especially noticeable in the presence of Arg. The change in the slope of the kinetic curve of Ph*b* aggregation in the presence of various additives indicates a change in the initial rate of aggregate growth (*v*_0_), which can be estimated using Equation (2). The value of *v*_0_ = 53,744 ± 2410 counts/s^2^ for Ph*b* increases in the presence of 0.05 mg/mL αB-Cr to the value of 73,109 ± 3550 counts/s^2^. Addition of 200 mM Bet leads to a decrease in the rate of aggregate growth to 32,726 ± 970 counts/s^2^, while the addition of 100 mM Arg increases the value of *v*_0_ to 111,051 ± 6460 counts/s^2^. This indicates that chemical chaperones significantly affect the rate of Ph*b* aggregation in the presence of αB-Cr at 48 °C and *IS* = 0.15 M. Figure 1B shows a protein control, where αB-Cr (0.05 mg/mL) and hen egg white lysozyme (0.05 mg/mL) were used as Ph*b* agent proteins. The experiment confirms that the inhibition of Ph*b* aggregation is the result of interaction with αB-Cr and does not occur in the presence of another protein. It should be noted that both agent proteins αB-Cr and lysozyme do not aggregate under the studied conditions.

To assess the effect of chemical chaperones on the αB-Cr adsorption capacity to the model protein, kinetic curves of Ph*b* aggregation were obtained in the presence of fixed concentrations of Bet (200 mM) or Arg (100 mM) and various concentrations of the heat shock protein (Figure 2A,B, respectively). The kinetic parameters of protein aggregation at the stage of nucleation (*K*_agg_, *t*_0_) and the stage of aggregate growth (*v*_0_, *t**) were calculated for each curve using Equations (1) and (2). The applicability of these equations for description of the kinetic curves of Ph*b* aggregation in the absence and in the presence of additives is shown in Appendix A.

Figure 3 shows the dependences of the relative values of the parameters characterizing the thermal aggregation of Ph*b* in the presence of αB-Cr and chemical chaperones on the ratio of molar concentrations of the heat shock protein and the target protein. Similar values obtained for Ph*b* in the presence of αB-Cr without additives were used as control values (black curves on all panels in Figure 3). Based on the *K*_agg_/*K*_agg,0_ and *v*_0_/*v*_0_^(0)^ dependences on [αB-crystallin]/[Ph*b*], the adsorption capacity (AC_0_) of αB-Cr with respect to Ph*b* can be determined at the stage of nucleation and at the stage of Ph*b* aggregates growth, respectively.

It was shown that 200 mM Bet or 100 mM Arg had almost no effect on the adsorption capacity of αB-Cr with respect to Ph*b* at the nucleation stage (Figure 3A). In the absence of chemical chaperones AC_0_ = 2.03 ± 0.08 (*R*^2^ = 0.983), in the presence of Bet 2.10 ± 0.17 (*R*^2^ = 0.924), and in the presence of Arg AC_0_ = 1.84 ± 0.15 (*R*^2^ = 0.923) Ph*b* monomers per 1 subunit of αB-Cr (Table 1).

At the same time, chemical chaperones have a noticeable effect on the value of *v*_0_ at the stage of Ph*b* aggregates growth (Figure 3C). In this case the AC_0_ value for αB-Cr without additives was 1.93 ± 0.10 (*R*^2^ = 0.967) Ph*b* monomer per 1 αB-Cr subunit, and it changed to 3.05 ± 0.34 (*R*^2^ = 0.863) or 0.82 ± 0.05 (*R*^2^ = 0.988) in the presence of 200 mM Bet or 100 mM Arg, respectively (Figure 3C, Table 1). This means that the adsorption capacity of the molecular chaperone increases under the action of Bet and decreases under the Arg influence.

As for the duration of the lag period, the *t*_0_ value increases more slowly in the presence of chemical chaperones but reaches higher values at high concentrations of αB-Cr, especially in the presence of Arg (Figure 3B, blue and red curves). The duration of the Ph*b* nucleation stage *t** in the presence of αB-Cr decreased upon addition of 200 mM Bet (Figure 3D, blue curve) and remained practically unchanged upon addition of 100 mM Arg (Figure 3D, red curve).

Thus, the main effect of chemical chaperones on the thermal aggregation of Ph*b* in the presence of αB-Cr at 48 °C is due to their influence on the stage of aggregate growth. Bet reduces the initial rate of Ph*b* aggregates growth and increases the adsorption capacity of αB-Cr to the target protein. On the contrary, Arg reduces the adsorption capacity and accelerates the formation of large aggregates. This means that chemical chaperones can significantly affect the PPIs of both HspB5 itself and its interaction with the target protein. At the nucleation stage, the contribution of chemical chaperones to the interaction between αB-Cr and Ph*b* is significantly less pronounced.

### 2.2. Effect of Chemical Chaperones on Phb Aggregation in the Presence of αB-Crystallin

To study the effect of Bet on the aggregation of the model protein in the presence of the protein chaperone, the kinetic curves of Ph*b* aggregation (0.3 mg/mL) were obtained at 48 °C in the presence of a constant concentration of αB-Cr (0.01 mg/mL) and various concentrations of Bet (Figure 4A). It was shown that the light scattering intensity (*I* − *I*_0_) decreases with increasing Bet concentration.

The calculation of the hydrodynamic radii (*R*_h_) of the formed aggregates showed that an increase in the Bet concentration leads to a slowdown in the formation of large Ph*b* aggregates in the presence of αB-Cr (Figure 4B). The initial point of the nucleation stage on the *R*_h_ curves on time at the moment *t* = *t*_0_, i.e., point with the coordinates {*t*_0_; *R*_h,0_} corresponds to the sizes of start aggregates *R*_h,0_ [37]. The *R*_h,0_ values can be determined from the dependence of the light scattering intensity on the hydrodynamic radius (Figure 4C), as described in [38]. The data obtained indicate that, upon the interaction of Ph*b* with αB-Cr (0.01 mg/mL) in the presence of Bet, the values of *R*_h,0_ increase from 30.8 to 52.5 nm with increasing Bet concentration up to 600 mM (Figure 4C, Table 2).

Analyzing the effect of Arg and αB-Cr on Ph*b* aggregation at 48 °C, the following points should be noted. As it was shown in our previous work [36], the size of Ph*b* start aggregates (*R*_h,0_) at *IS* = 0.15 M remained unchanged, regardless of the Arg concentration. In the presence of 0.01 mg/mL αB-Cr and Arg, the *R*_h,0_ sizes of Ph*b* were found to be the same as in the absence of Arg (Table 2; 30.8 ± 2) and were equal to 31.9 ± 2 nm (data not shown). Thus, it can be concluded that αB-Cr reduces the size of Ph*b* start aggregates during thermal aggregation at 48 °C. The addition of Bet leads to an increase in the *R*_h,0_ sizes, while Arg does not affect the size of the start aggregates.

### 2.3. The Effect of Chemical Chaperones on Thermal Stability of αB-Crystallin

The use of differential scanning calorimetry (DSC) makes it possible to assess how chemical chaperones affect the thermal stability of the studied proteins. Figure 5 shows the DSC profiles for αB-Cr (1.0 mg/mL) in the absence of any additives (black curve) and in the presence of 100 mM Arg (red curve), 200 mM Bet, and 500 mM Bet (green and blue curves, respectively) at *IS* = 0.15 M.

According to DSC data, Arg slightly destabilizes the tertiary structure of αB-Cr (1 mg/mL), shifting its DSC profile towards lower temperatures (Figure 5, black and red curves, respectively). The temperature of the thermal transition maximum (*T*_max_) for αB-Cr changes from 60.5 °C in the absence of additives to 60.1 °C in the presence of 100 mM Arg (Table 3). In contrast with Arg, the presence of 200 mM or 500 mM Bet shifts the DSC profiles of the protein towards higher temperatures up to *T*_max_ = 61.6 or 62 °C, respectively (Figure 5, green and blue curves; Table 3). This indicates the αB-Cr stabilization under the action of Bet. It should be noted that the presence of these chemical chaperones has practically no effect on the calorimetric enthalpy of αB-Cr thermal transition (Table 3, Δ*H*_cal_). This means that the structural rearrangements in the protein molecule induced by Bet or Arg are insignificant.

A similar effect of these chemical chaperones on the protein tertiary structure, demonstrating its stabilization in the presence of Bet and destabilization under the influence of Arg, was also shown for Ph*b*. These data were presented in our previous works [35,36], and therefore, such experiments were not repeated within the framework of the current one.

### 2.4. Analytical Ultracentrifugation of αB-Crystallin, Phb, and Their Mixture in the Presence of Chemical Chaperones under Heat Shock Conditions

Studies of the kinetics of Ph*b* thermal aggregation in the presence of αB-Cr and Bet or Arg have shown that chemical chaperones mainly affect the stage of aggregate growth. Thus, it was interesting to study their effect on the oligomeric state of the protein chaperone and Ph*b*, as well as on their interaction for a longer time, namely, up to 3 h of incubation at 48 °C. For this purpose, the analytical ultracentrifugation (AUC) method was used.

It is well known that the dynamic quaternary structure of αB-Cr is rearranged with a change in temperature and that this is a rather long process. Therefore, all samples and controls were studied simultaneously under the same conditions. Figure 6 shows the effect of Bet on the sedimentation behavior of αB-Cr at 48 °C. The *c*(*s*) distribution for αB-Cr exhibits one main peak with sedimentation coefficient (*s*_20,w_) 20.4 ± 0.3 S and several minor peaks with *s*_20,w_ (18.6 ± 0.6 S; 22.8 ± 0.8 S; 31.2 ± 0.9 S). Comparison of the *c*(*s*) distributions in Figure 6 for αB-Cr in the absence of Bet (black line) with others shows that in the presence of 100 mM or 300 mM Bet, the main oligomeric form of αB-Cr (peak with *s*_20,w_ = 20.4 ± 0.3 S) dissociates to form smaller ensembles (peak with *s*_20,w_ = 16.5 ± 0.2 S). Under given conditions, the molecular mass of αB-Cr determined by the Svedberg equation (see Section 4.6) was equal to ~809 kDa (Appendix A), and the molecular mass of αB-Cr in the presence of 300 mM Bet was equal to 486 kDa. Taking the molecular mass of αB-Cr monomer equal to 20 kDa, one can estimate that under these conditions major peak on *c*(*s*) distribution for αB-Cr (Figure 6) corresponds to 40-mer, while the major peak for αB-Cr in the presence of 300 mM Bet (Figure 6) corresponds to approximately 24-mer. When comparing the initial distribution for αB-Cr in the absence of Bet with the distribution in the presence of 500 mM Bet, the major peak of the distribution (20.5 S) almost coincides with the initial peak of αB-Cr (20.4 S). The results obtained indicate that low concentrations of Bet (up to 300 mM) stimulate dissociation of αB-Cr, while higher concentrations prevent this. The effect of higher Bet concentration (500 mM) on the oligomeric state of the αB-Cr at room temperature is given in Appendix A. The all *c*(*s*) distribution is shifted slightly towards larger values of the sedimentation coefficient in the presence of 500 mM Bet. This indicates that the portion of larger oligomers in the *c*(*s*) distribution in the presence of 500 mM Bet increases (Appendix A). However, the weight-average sedimentation coefficients for both distributions are equal.

Figure 7 shows that under the studied conditions, Ph*b* is represented by an unfolded monomeric form (peak with *s*_20,w_ = 4.5 ± 0.4 S) and a partly unfolded dimeric form (8.5 ± 0.3 S, black dotted line). In the presence of 500 mM Bet, the *c*(*s*) distribution for Ph*b* exhibits a broad peak (8.3 ± 0.8 S) with small shoulders at 10 ± 0.3 S and 6.7 ± 0.4 S (blue line). This means that 500 mM Bet can stabilize the native dimer of Ph*b* (10 S), denatured dimer (8.3 S), and small oligomers. An analysis of the *c*(*s*) distributions in Figure 7 allows us to conclude that in the (Ph*b* + αB-Cr) mixture (green line), the main (peak 8.3 ± 0.7 S) and minor (10 ± 0.4 S) forms are also stabilized. If we consider the *c*(*s*) distribution for a mixture of Ph*b* (0.4 mg/mL) in the presence of αB-Cr (0.2 mg/mL) and 500 mM Bet (black solid line), we can see that the major peak of αB-Cr (with *s*_20,w_ = 20.4 ± 0.3 S; red line) disappears and that the fraction of the main form with a peak of 8.3 S (black solid line) increases (the area under the curve increases), which means that 500 mM Bet enhances the interaction of αB-Cr with Ph*b* under the studied conditions. In addition, many different in size Ph*b* complexes with αB-Cr are presented (with *s*_20,w_ 6.3 ± 0.4, 13 ± 0.5, 16 ± 0.7, 17.8 ± 0.6, 24 ± 0.5, 27 ± 0.5 S).

Figure 8 shows that the *c*(*s*) distribution for αB-Cr in the presence of 100 mM Arg is shifted toward smaller sedimentation coefficients. This suggests that the fraction of smaller oligomers increases in the presence of 100 mM Arg. Unfortunately, it was not possible to observe the effect of Arg on the interaction of αB-Cr with Ph*b* at 48 °C, since the samples precipitated.

The effect of Bet and Arg on the aggregation of Ph*b* and the interaction of Ph*b* with αB-Cr was studied at 20 °C after the protein samples were preheated at 48 °C for 3 h and cooled to 20 °C. The data obtained by AUC are given in Table 4.

## 3. Discussion

PPIs include not only interactions of a chaperone with a target protein (in this case, αB-Cr with Ph*b*) but also chaperone’s polydisperse assemblies of subunit-exchanging oligomers. Additives that act on both protein chaperone and target protein will affect these PPIs.

In this work, we have shown that chemical chaperones can influence the conformation of αB-Cr subunits and interaction both with each other and with the target protein, which results in a change in the adsorption capacity of αB-Cr with respect to Ph*b*. According to DSC data, Bet stabilizes both proteins ([36]; Figure 5, green and blue curves), while Arg destabilizes them ([35]; Figure 5, red curve). Stabilization of Ph*b* by Bet and destabilization of the protein by Arg is also observed in AUC experiments (Table 4). The data on the effect of Bet or Arg on the aggregation of Ph*b* and the interaction of Ph*b* with αB-Cr after the protein samples were preheated at 48 °C for 3 h and cooled to 20 °C showed that Bet inhibited formation and precipitation of higher-order aggregates of Ph*b* and decreased significantly (by 5 fold) fraction γ_agg_ for mixture Ph*b* + αB-Cr (Table 4). Thus, an enhancement in the interaction between αB-Cr and the target protein in the presence of Bet leads to an increase in the adsorption capacity of the chaperone with respect to Ph*b* (AC_0_) in the presence of Bet. Arg, on the contrary, stimulates the precipitation both Ph*b* and the mixture Ph*b* + αB-Cr (Table 4) and decreases AC_0_ value.

The initial binding stoichiometry chaperone–client protein in the presence of 200 mM Bet increases by 1.5 times (from 1:2 to 1:3), and in the presence of 100 mM Arg decreases by 2.4 times (from 1:2 to 1:0.82; Figure 3C, Table 1) on the stage of aggregate growth. This means that the anti-aggregation activity of HspB5 towards the target protein increases in the presence of Bet and decreases in the presence of Arg, which results in a change in the kinetics of Ph*b* thermal aggregation at 48 °C and ionic strength of 0.15 M (Figure 1A, green and blue curves).

The influence of chemical chaperones on the quaternary structure of HspB5, and hence on PPIs, was also shown by the AUC method (Figure 6 and Figure 8). The addition of Bet up to 300 mM or 100 mM Arg leads to disassembly of HspB5 and reorganization of the αB-Cr subunits in chaperone assembly (Figure 6 and Figure 8). With an increase in Bet concentration up to 500 mM, the excluded volume effect prevents the dissociation of the molecular chaperone (Figure 6). Earlier we showed a significant increase in the size of αB-Cr at 48 °C under crowding conditions arising from the presence of 1 M TMAO and/or other crowding agents [39]. The estimated number of subunits in the αB-Cr oligomer increased from 22-mers up to 50-mers under conditions of mixed crowding created by 1M TMAO + PEG [34,39]. Grosas and colleagues also reported an increase in the size of αB-Cr under crowding conditions [40]. It should be emphasized that the presence of the target protein stimulated the dissociation of large chaperone ensembles and the formation of chaperone-client complexes with different sizes even under crowding conditions, which increased the sizes of sHsps [34,39,41]. The AUC data obtained in the present work for a mixture of αB-Cr and Ph*b* in the presence of 500 mM Bet confirm this. On the *c*(*s*) distribution for HspB5 + Ph*b* under given conditions, the peak (20.4 S) corresponding to free αB-Cr disappears, and peaks with lower sedimentation coefficients appear, corresponding to smaller oligomeric complexes of chaperone-Ph*b*. This is consistent with the findings of Schiedt et al. [7]. Based on the thermodynamic and kinetic characteristics of the binding of αB-Cr to α-synuclein fibrils, the authors concluded that there was a step of chaperone activation through the disassembly of chaperone complexes [7]. This mechanism is consistent with previous findings on substrate activation and disassembly of other sHsps with anti-aggregation activity [41,42,43].

In the presence of a constant αB-Cr concentration (0.01 mg/mL), Ph*b* aggregation slows down with an increase in Bet concentration (Figure 4A). However, the sizes of the start aggregates (*R*_h,0_) characterizing the initial point of Ph*b* aggregation increase from 30.8 to 52.5 nm (Table 2), with an increase in the concentration of Bet from 0 to 600 mM in a mixture of Ph*b* + αB-Cr. Crowding greatly affects PPIs and the sizes of start aggregates. For aggregation of alpha-lactalbumin denatured with DTT, a significant increase in the size of start aggregates (*R*_h,0_) was shown in the presence of crowding agents. For example, the parameter *R*_h,0_ increased 5.7 times in the presence of PEG (50 mg/mL) [44]. It was assumed that when unfolded Ph*b* molecules form complexes with αB-Cr, the centers necessary for the formation of start aggregates were closed. Their formation occurred due to additional contacts in these complexes. This required additional time. Therefore, the duration of the lag period and the nucleation stage increased [45]. The binding of Bet to αB-Cr led to the release of a part of the centers necessary for binding the unfolded Ph*b* molecules to form start aggregates. As the concentration of Bet increased, more and more of these centers were released, and the sizes of the start aggregates approached those for Ph*b* in the absence of all additives (Table 2). In addition, the crowding arising from the presence of high concentration of Bet can promote the formation of Ph*b*–HspB5 complexes with a more compact conformation as well as the complexes with larger sizes. However, further growth of aggregates slows down with an increase in Bet concentration (Figure 4B).

In the presence of 0.01 mg/mL αB-Cr and Arg at 48 °C and ionic strength of 0.15 M, the *R*_h,0_ sizes of Ph*b* were found to be the same as in the absence of Arg and were equal to 31.9 ± 2 nm. As it was shown in our previous work [35], the start aggregates size (*R*_h,0_) of Ph*b* at 48 °C and *IS* = 0.15 M in the absence of αB-Cr remained unchanged, regardless of the concentration of Arg. Thus, the action of Arg does not affect the size of the start aggregates either in the absence or in the presence of αB-Cr.

It should be noted that the binding of Arg to Ph*b* causes changes in conformation of the enzyme [35]. The bond between monomers in the dimer weakens, and the dimer breaks down into monomers, which quickly aggregate. Obviously, αB-Cr cannot significantly affect this process, since almost all of the protein precipitates both in the presence and absence of αB-Cr (Table 4).

Srinivas [28] showed that there was little effect of Arg on the secondary or tertiary structure of α-crystallin but that Arg mediated an increase in subunit exchange and destabilization of the α-crystallin structure. In some cases, such destabilization led to an increase in the activity of the chaperone [27,28,29,30], while in others, as in the case of catalase aggregation at 55 °C, 100 mM Arg decreased the ability of αB-Cr to inhibit aggregation of the target protein [29].

Chemical chaperones can act at different stages of the protein aggregation process. In this work, Bet and Arg affected the stage of Ph*b* aggregate growth in the presence of αB-Cr. Another chemical chaperone, trehalose, acts on the nucleation stage of thermal aggregation of Ph*b* in the presence of αB-Cr [46]. According to our data obtained for αB-Cr and another target protein (UV-irradiated Ph*b*), Arg acts on both stages of UV-Ph*b* aggregation (S3).

Bet stabilizes Ph*b* [36] and αB-Cr (Figure 5, green and blue curves), and the anti-aggregation activity of the chaperone increases (Figure 3C, blue curve). However, in some cases, the stabilization of the chaperone results in a decrease in its activity. The major protein of horse seminal plasma, HSP-1/2, exhibits membranolytic and chaperone-like activities. Addition of L-carnitine increases thermal stability but decreases both chaperone-like and membranolytic activities of this protein [47]. Moreover, this proves that the effect of protein chaperones is target protein-specific. It can be assumed that an enhancement of the anti-aggregation activity of αB-Cr towards Ph*b* in the presence of Bet is associated with the stabilization of Ph*b,* αB-Cr, and the complexes of the chaperone with the target protein and that a decrease in the presence of Arg is due to the destabilization of them. Thus, chemical chaperones have a strong effect on PPIs, both on Ph*b*, αB-Cr, and on their complexes.

However, McHaourab et al. suggest [48], and subsequent work confirmed this [7,49], that αB-Cr has various binding sites with high and low affinity to protein substrate. Some of binding sites on αB-Cr are favored by phosphorylation or interaction with compounds such as Arg, whilst others are not affected [29]. The chaperone itself may be in a low or high affinity state. Under physiological conditions, the low affinity state is predominant, and stress situation leads to the high affinity one [49]. This shift between affinity states is commonly regulated by the composition of oligomeric species. Different triggers reversibly change not only the oligomer equilibrium but also the chaperone concentration, which increases under stress conditions (such as high temperatures and other substances present in a cell) and regulates the activity of sHsps and PPIs chaperone with the target protein [49,50].

The activity of protein chaperones, including αB-Cr, can be regulated by chemical modification [51] or post-translational modification [52]. In addition, chemical chaperones can also control the structure and function of αB-Cr, stabilizing or destabilizing its structure and enhancing or weakening the function of the chaperone. We can assume that the different effect of Bet and Arg on AC_0_ of αB-Cr with respect to Ph*b* may be associated with allosteric changes caused by them in the substrate-binding sites of αB-Cr, which increase or decrease its affinity to the unfolded substrate, Ph*b*. These interactions are very helpful in understanding the details of the structural changes and chaperone function of αB-Cr.

To effectively refold heat-denatured proteins, sHsps must coordinate with other chaperone families [53]. sHsps create a reservoir of partly unfolded client proteins, which then are refolded by other ATP-dependent chaperones. Under stress conditions, sHsps bind non-native proteins, prevent their irreversible aggregation, and hold them in a refoldable state. Upon restoration of physiological conditions, the non-native protein will either dissociate from the complex spontaneously, or ATP-dependent chaperones such as Hsp70 will release the non-native polypeptide [54]. The mechanism of sHsps action with conjunction with Hsp70 system is universal among eukaryotes and prokaryotes, and it suggests that sHsps may not physically interact with Hsp70 [55].

Understanding chaperone–client interactions helps find substances that can regulate the activity of sHsps to create therapies for treatment of the wide range of diseases associated with protein misfolding and aggregation. Thus, our studies contribute to understanding the mechanism of interaction between chaperones and proteins.

## 4. Materials and Methods

### 4.1. Materials

Hepes, arginine hydrochloride, betaine, and hen egg white lysozyme were obtained from “Sigma” (St. Louis, MO, USA). 1,4-Dithiothreitol (DTT) was purchased from “Panreac” (Barcelona, Spain). NaCl was obtained from “Reakhim” (Moscow, Russia). The procedure for isolation of Ph*b* from rabbit skeletal muscle was described in [56]. Lysozyme diluted in 0.03 M Hepes buffer pH 6.8, containing 0.15 M NaCl, was centrifugated for 15 min at 12,850× *g* at 6 °C.

### 4.2. Isolation of αB-Crystallin (HspB5)

The human HspB5 coding sequence was cloned into the pET23 vector for expression in *E. coli* cells as described in [57]. Overexpressed HspB5 was purified by salting out with ammonium sulfate followed by the gel filtration. The purest fractions (according to SDS gel electrophoresis) were combined, concentrated, aliquoted, and stored at −80 °C.

### 4.3. Dynamic Light Scattering (DLS) Study

A commercial Photocor Complex instrument (Photocor Instruments Inc., College Park, MD, USA) was used to measure light scattering intensity. He-Ne laser (Coherent, Santa Clara, CA, USA, Model 31-2082, 632.8 nm, 10 mV) was used as the light source. The DYNALS computer program (Alango, Tirat Carmel, Israel) was used for polydisperse analysis of dynamic light scattering data. The hydrodynamic radius (*R*_h_) of protein aggregates was calculated as described in [58] using the values of refractive indexes and dynamic viscosity, which are given in Appendix A. The kinetics of Ph*b* aggregation at 48 °C were studied in 0.03 M Hepes buffer, pH 6.8, containing 0.5 mM DTT with *IS* = 0.15 M adjusted by NaCl, where necessary. The buffer was placed in a cylindrical cell with an inner diameter of 6.3 mm and incubated for 5 min at 48 °C. To study the effect of additives and their mixtures, these substances were incubated in the cell for 5 min at 48 °C before adding the target protein. When studying the kinetics of Ph*b* aggregation, scattering light was collected at a scattering angle of 90°.

### 4.4. Determination of the Adsorption Capacity of the Chaperone at Different Stages of Target Protein Aggregation

The anti-aggregation activity of the protein chaperone αB-Cr can be characterized by the value of its initial adsorption capacity (AC_0_), which shows how many target protein molecules are bound by 1 chaperone molecule. The nucleation stage is described by the following equation:(1)$$I-I_{0}=K_{\mathrm{agg}}{\left(t-t_{0}\right)}^{2},\;(t>t_{0})$$
where *I* is the light scattering intensity, *t* is the time, *I*_0_ is the initial value of the light scattering intensity at *t* = 0, and *t*_0_ is the time instant at which the initial increment of the light scattering intensity is registered, i.e., lag-period on kinetic curves of Ph*b* aggregation, *K*_agg_ is a parameter characterizing the acceleration of the aggregation at the stage of nucleation [59,60].

To describe the initial part of the stage of protein aggregates growth, the following equation was used [30]:(2)$$I-I_{0}=v_{0}\left(t-t*\right)-B{\left(t-t*\right)}^{2},\;(t>t*)$$
where, *t** is the duration of the nucleation stage, determined by the segment on the abscissa axis, cut off by the theoretical curve calculated from the Equation (2) at *I* = 0, *v*_0_ is the initial rate of aggregation, and B is a constant.

At the nucleation stage, the adsorption capacity of the chaperone AC_0_ can be defined as the reciprocal value of the segment cut off on the abscissa axis by the initial linear part of the *K*_agg_/*K*_agg,0_ dependence on the [αB-crystallin]/[Ph*b*] ratio. The parameters *K*_agg_ and *K*_agg,0_ are parameters characterizing the acceleration of the aggregation at the stage of nucleation in the presence and in the absence of a chemical chaperone, respectively. [αB-crystallin]/[Ph*b*] is the ratio of the molar concentration of αB-Cr, calculated for a protein subunit with a molecular weight of 20 kDa, to the molar concentration of Ph*b*, calculated for a monomer with a molecular weight of 97.4 kDa.

At the stage of protein aggregate growth, the initial adsorption capacity of the chaperone AC_0_ can be determined in a similar way from the initial linear part of the *v*_0_/*v*_0_^(0)^ dependence on the [αB-crystallin]/[Ph*b*] ratio. Parameters *v*_0_ and *v*_0_^(0)^ characterize the initial rate of the aggregation at the stage of aggregate growth in the presence and in the absence of a chemical chaperone, respectively.

### 4.5. Differential Scanning Calorimetry (DSC) Studies

Differential scanning calorimetry was used to investigate the thermal denaturation of αB-Cr in the absence or in the presence of Bet and Arg. The experiments were performed on a MicroCal VP-Capillary DSC differential scanning calorimeter (Malvern Instruments, Northampton, MA 01060, USA) at a heating rate of 1 °C/min in 0.03 M Hepes buffer, pH 6.8. The ionic strength in all experiments was 0.15 M. The αB-crystallin concentration was 1 mg/mL. In the experiments with chemical chaperones, the same concentration of chaperone was added to both control and sample cells. The correction of the calorimetric traces, analysis of the temperature dependence of excess heat capacity, the thermal stability estimation, and calculation of calorimetric enthalpy (Δ*H*_cal_) was performed as described in [30].

### 4.6. Analytical Ultracentrifugation (AUC) 

Sedimentation velocity (SV) experiments were conducted at 48 °C in a model E analytical ultracentrifuge (Beckman Instruments, Palo Alto, CA, USA), equipped with absorbance optics, a photoelectric scanner, a monochromator, and a computer on-line. A four-hole rotor (An-F Ti) and 12 mm double sector cells were used in the experiments. The rotor was preheated in the thermostat overnight before the run at 48 °C. Sedimentation profiles of samples in a 0.03 M Hepes buffer, pH 6.8 with *IS* = 0.15 M (adjusted by NaCl, where necessary), were recorded by measuring the optical density at 280 nm. All cells were scanned simultaneously with a 2.5 min interval. Differential sedimentation coefficient distributions [*c*(*s*) vs. *s*] were determined and corrected to the standard conditions (a solvent with the density and viscosity of water at 20 °C) using SEDFIT program [61]. The *c*(*s*) analysis was performed with regularization at confidence levels of 0.68 and a floating frictional ratio, time-independent noise, baseline, and meniscus position.

To estimate the oligomeric state and molecular mass of αB-Cr in the absence and in the presence of Bet, the Svedberg equation was used: *M* = *sRT*/*D*(1 − *νρ*),(3)
where *ν* is the partial specific volume of a protein, *ρ* is solution density, *R* is molar gas constant, *T* is temperature in Kelvin, *s* is a sedimentation coefficient, *D* is a diffusion coefficient. For calculations we used sedimentation coefficients (*s*), determined by AUC at 48 °C, and diffusion coefficients (*D*), determined by DLS at 48 °C.

The values of density and dynamic viscosity of the solutions used in the AUC measurements at 48 °C are presented in Appendix A [62].

## 5. Conclusions

In this work, we have shown that chemical chaperones can influence the tertiary and quaternary structure of both the target protein and the protein chaperone. Bet stabilizes both Ph*b* and αB-Cr, increasing the anti-aggregation activity of αB-Cr. Arg, on the contrary, reduces the stability of both proteins and reduces the anti-aggregation activity of αB-Cr. The presence of Ph*b*, in turn, also affects the quaternary structure of αB-Cr, causing its disassembly. Since the dynamic quaternary structure of αB-Cr is inextricably linked with its anti-aggregation activity, any changes in the structure of the protein chaperone affect its PPI with the substrate protein.

## Figures and Tables

**Figure 1 ijms-23-03816-f001:**
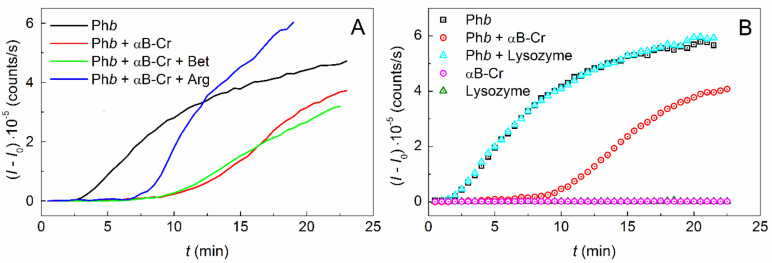
Effect of chaperones on the kinetics of Ph*b* aggregation at 48 °C. (**A**) The dependences of the light scattering intensity (*I* − *I*_0_) on time obtained for Ph*b* (0.3 mg/mL, black curve) and for Ph*b* in the presence of αB-Cr (0.05 mg/mL, red curve) and its mixture with 200 mM Bet (green curve) or 100 mM Arg (blue curve). (**B**) Negative protein control. The aggregation curves for Ph*b* (0.3 mg/mL, black squares), αB-Cr (0.05 mg/mL, magenta circles), hen egg white lysozyme (0.05 mg/mL, olive triangles), and mixtures Ph*b* + αB-Cr (red circles) and Ph*b* + lysozyme (cyan triangles).

**Figure 2 ijms-23-03816-f002:**
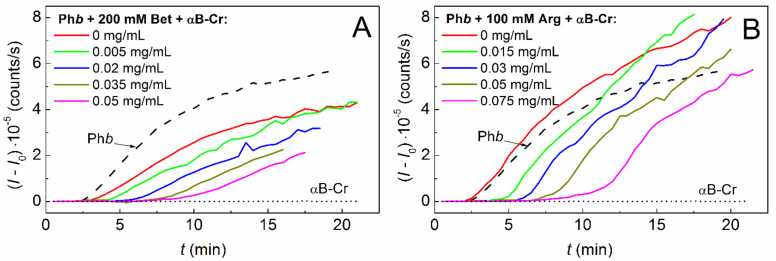
Effect of αB-Cr on the aggregation kinetics of Ph*b* at 48 °C in the presence of chemical chaperones. The dependences of the light scattering intensity (*I* − *I*_0_) on time obtained for Ph*b* (0.3 mg/mL) in the presence of various concentrations of αB-Cr and (**A**) 200 mM Bet or (**B**) 100 mM Arg. αB-Cr concentrations are indicated on the panels. Dash curves and short dash curves on the panels correspond to Ph*b* and αB-Cr in the absence of any additives, respectively.

**Figure 3 ijms-23-03816-f003:**
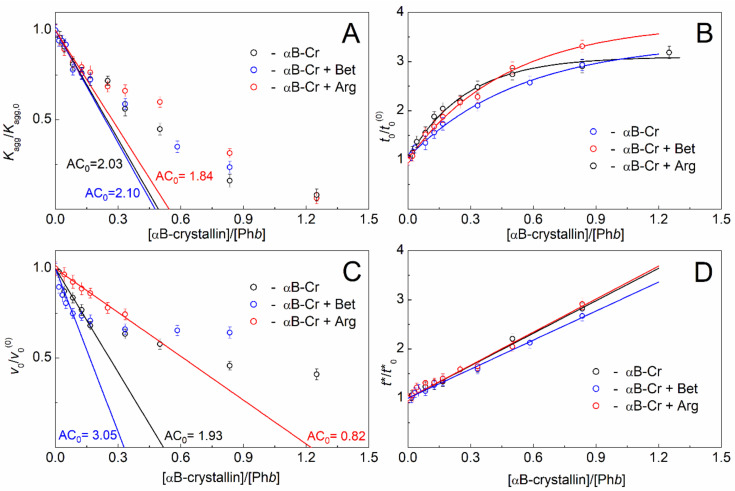
Effect of αB-Cr on the main kinetic parameters of Ph*b* aggregation (0.3 mg/mL) at 48 °C in the absence of additives, in the presence of 200 mM Bet, and in the presence of 100 mM Arg. (**A**) Dependences of the relative acceleration of Ph*b* aggregation at the nucleation stage (*K*_agg_/*K*_agg,0_), (**B**) the relative values of the lag periods on the kinetic curves of Ph*b* aggregation (*t*_0_/*t*_0_^(0)^), (**C**) the relative initial rate of Ph*b* aggregation (*v*_0_/*v*_0_^(0)^), and (**D**) the relative values of the nucleation stage duration (*t**/*t**_0_) on the ratio of molar concentrations of αB-Cr and Ph*b*. Black, blue, and red curves on all panels were obtained in the absence of additives, in the presence of 200 mM Bet, and in the presence of 100 mM Arg, respectively. The error bars were calculated using three independent measurements.

**Figure 4 ijms-23-03816-f004:**
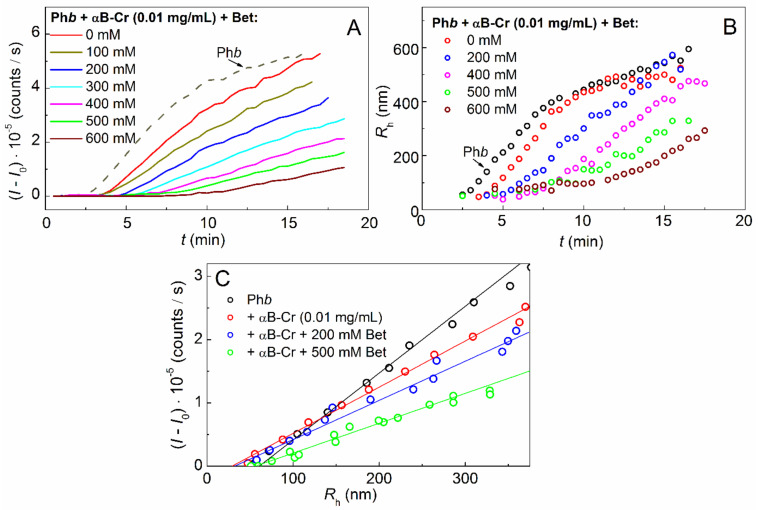
Effect of Bet on the kinetics of Ph*b* aggregation (0.3 mg/mL) in the presence of 0.01 mg/mL αB-Cr at 48 °C. The dependences of (**A**) the light scattering intensity (*I* − *I*_0_) and (**B**) the hydrodynamic radius of protein aggregates (*R*_h_) on time obtained at various concentration of Bet which are shown on the panels. (**C**) Initial parts of the dependences of (*I* − *I*_0_) on (*R*_h_). The dashed curve on panel A and black circles on panels B and C correspond to Ph*b* without any additives.

**Figure 5 ijms-23-03816-f005:**
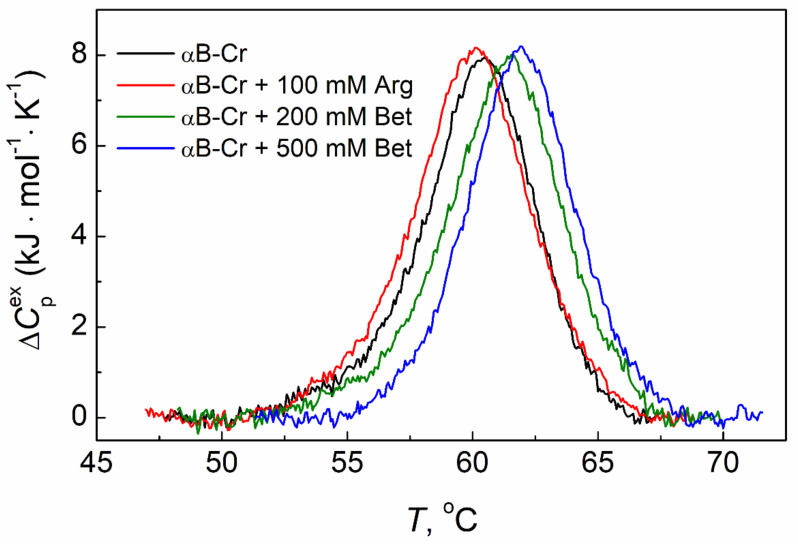
The effect of Bet or Arg on the thermal stability of αB-Cr. The DSC profiles for αB-Cr (1 mg/mL) in the absence of additives and in the presence of 100 mM Arg, 200 mM Bet, and 500 mM Bet (black, red, green, and blue curves, respectively). *IS* = 0.15 M. The heating rate was 1 °C/min.

**Figure 6 ijms-23-03816-f006:**
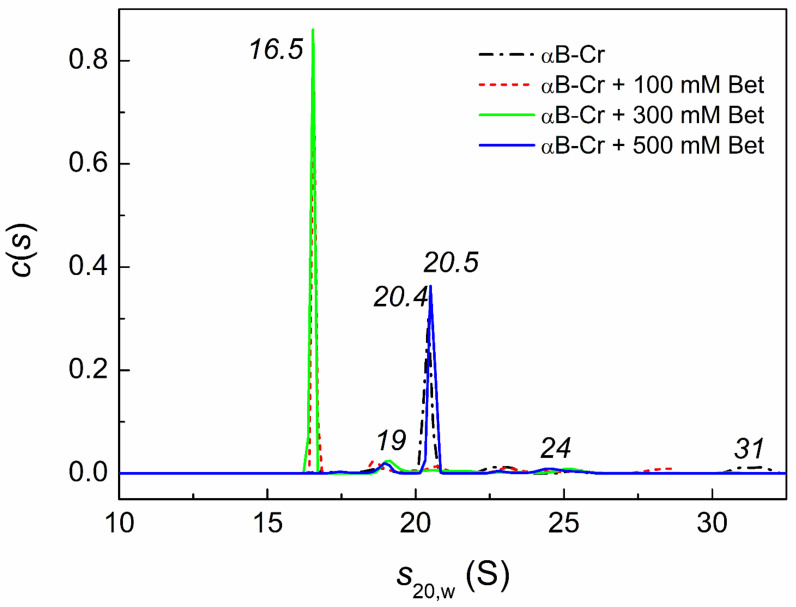
Effect of Bet on the oligomeric state of αB-Cr under heat stress conditions (48 °C). Differential sedimentation coefficient distributions, *c*(*s*), for αB-Cr (0.2 mg/mL) in the presence of various concentrations of Bet: 0 (black dash dotted line), 100 mM (red dashed line), 300 mM (green solid line), 500 mM (blue solid line). The values of sedimentation coefficients are shown on the panel in italics. The *c*(*s*) distributions were obtained at 48 °C and transformed to standard conditions. Rotor speed was 48,000 rpm. Total time at 48 °C was 80 min.

**Figure 7 ijms-23-03816-f007:**
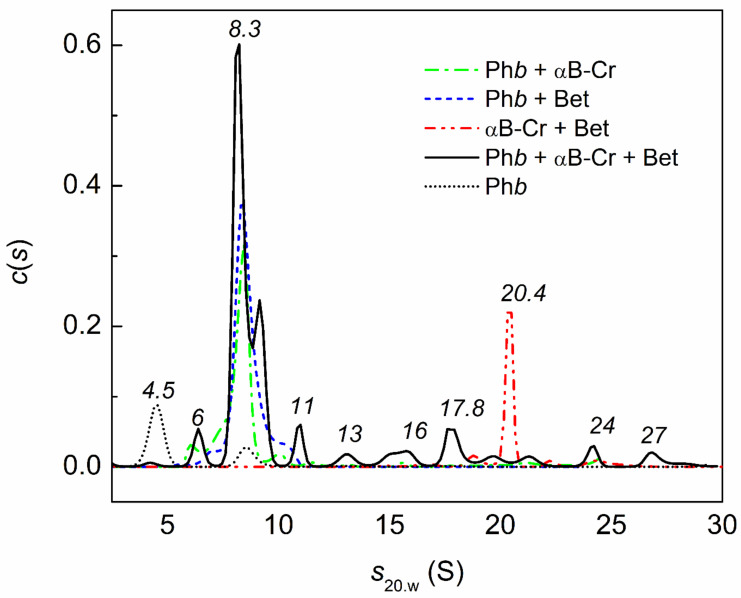
Effect of Bet on the interaction of αB-Cr with Ph*b* at 48 °C. The *c*(*s*) distributions for Ph*b* (0.4 mg/mL) in the presence of αB-Cr (0.2 mg/mL) and 500 mM Bet (black solid line); for Ph*b* in the presence of αB-Cr (green dash dotted line); for Ph*b* in the presence of 500 mM Bet (blue short dashed line); for Ph*b* (black dotted line); for αB-Cr in the presence of 500 mM Bet (red dash dot dotted line). The values of sedimentation coefficients are shown on the panel in italics. The *c*(*s*) distributions were obtained at 48 °C and transformed to standard conditions. Rotor speed was 48,000 rpm. Total time at 48 °C was 140 min.

**Figure 8 ijms-23-03816-f008:**
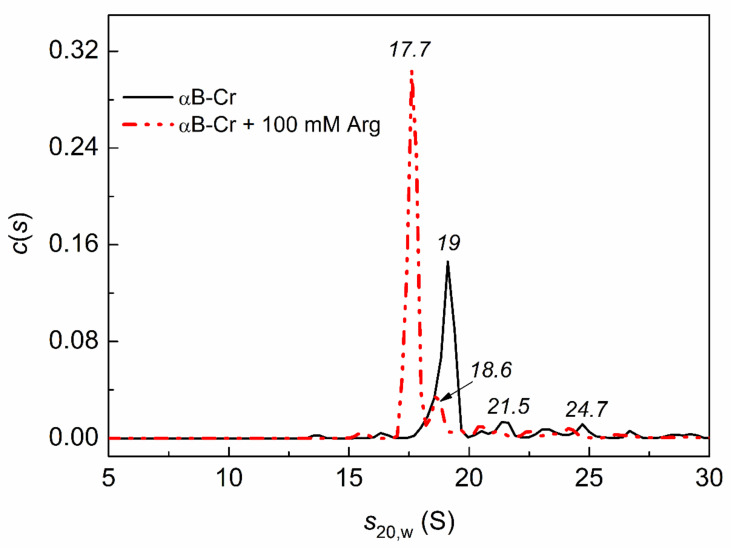
Effect of Arg on the oligomeric state of αB-Cr under heat stress conditions (48 °C). The *c*(*s*) distributions for αB-Cr (0.2 mg/mL) in the absence (black solid line) and in the presence of 100 mM Arg (red dash dot dotted line). The values of sedimentation coefficients are shown on the panel in italics. The *c*(*s*) distributions were obtained at 48 °C and transformed to standard conditions. Rotor speed was 48,000 rpm. Total time at 48 °C was 210 min.

**Table 1 ijms-23-03816-t001:** Effects of Bet or Arg on the adsorption capacity of αB-Cr to Ph*b* at 48 °C and *IS* = 0.15 M. The table shows the average data obtained from three experiments.

Component with Variable Concentration	Nucleation Stage	Stage of Aggregate Growth
αB-Cr	AC_0_ = 2.03 ± 0.08 Ph*b* monomer per 1 subunit of αB-Cr	AC_0_ = 1.93 ± 0.10 Ph*b* monomer per 1 subunit of αB-Cr
αB-Cr in the presence of 200 mM Bet	AC_0_ = 2.10 ± 0.17 Ph*b* monomer per 1 subunit of αB-Cr	AC_0_ = 3.05 ± 0.34 Ph*b* monomer per 1 subunit of αB-Cr
αB-Cr in the presence of 100 mM Arg	AC_0_ = 1.84 ± 0.15 Ph*b* monomer per 1 subunit of αB-Cr	AC_0_ = 0.82 ± 0.05 Ph*b* monomer per 1 subunit of αB-Cr

**Table 2 ijms-23-03816-t002:** The values of *R*_h__,0_ for Ph*b* aggregates in the presence of αB-Cr (0.01 mg/mL) and different concentrations of Bet at 48 °C and *IS* = 0.15 M. The table shows the average data obtained from three experiments.

Sample	Betaine (mM)	*R*_h__,0_ (nm)
Ph*b* (0.3 mg/mL)	0	62.7 ± 3
Ph*b* + αB-Cr	0	30.8 ± 2
	200	34.1 ± 2
	400	38.1 ± 2
	500	48.8 ± 2
	600	52.5 ± 2

**Table 3 ijms-23-03816-t003:** Calorimetric parameters obtained from the DSC data for the thermal transitions of αB-Cr (1 mg/mL) in the absence and in the presence of chemical chaperones.

Sample	*T*_max_ (°C)	Δ*H*_cal_ (kJ·mol^−1^)
αB-Cr	60.5 ± 0.1	44.2 ± 2.7
αB-Cr + 100 mM Arg	60.1 ± 0.1	48.9 ± 2.9
αB-Cr + 200 mM Bet	61.6 ± 0.1	46.4 ± 2.8
αB-Cr + 500 mM Bet	62.0 ± 0.1	44.4 ± 2.7

**Table 4 ijms-23-03816-t004:** Estimation of the fraction of aggregated protein (γ_agg_) precipitated during the acceleration of the rotor in the AUC experiment at 20 °C. Rotor speed was 48,000 rpm. Before the experiment, the samples were heated at 48 °C for 3 h and then quickly (2 min) cooled.

Sample	γ_agg_ (%)
Ph*b* (0.5 mg/mL)	33
Ph*b* + αB-Cr + 500 mM Bet	12
Ph*b* + 500 mM Bet	0
Ph*b* + 100 mM Arg	96
Ph*b* + αB-Cr + 100 mM Arg	96

## Data Availability

The study did not report any publicly archived data.

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
