# Peer review of "Effect of Betaine and Arginine on Interaction of αB-Crystallin with Glycogen Phosphorylase b"

_ijms, 2022, doi:10.3390/ijms23073816_

Round 1

Reviewer 1 Report

This manuscript describes the study of the aggregation behaviour of aB-Cr in the presence of various chaperons. The authors show that Bet and Arg can influence the conformation of aB-Cr and its interaction with target proteins using a variety of experiments. However, each experiment is not well introduced in terms of stating why this was carried out and what the results show in the context of the behaviour of the protein. There are never any comparisons made with other systems so it is hard to know whether the changes seen are surprising/significant or to be expected. Why were the conditions used chosen, why 48oC and why this ionic strength for example. The links to the overall function of the protein are also not very clear. This manuscript is only really of relevance to others working directly on this system.

Author Response

Dear reviewer,

Thank you very much for your comments. Our reply is in the file.

Reviewer 2 Report

In this work the authors characterize the effect the holdase chaperone É‘B-crystallin and the two additives betaine and arginine have on the thermal denaturation and aggregation behavior of the enzyme glycogen phosphorylase b.

The authors provide sufficient information and details in introduction and discussion and put their results into context with the literature.

It is known that É‘B-crystallin functions as a holdase chaperone to prevent aggregation under stress condition and when the stress is relieved it has the ability to pass the client protein onto the Hsp70 refolding machinery. An interesting question that was not discussed is if the effect of betaine and É‘B-crystallin on the aggregation of Phb affect this process. An experiment that measures enzyme activity after aggregation in the absence and presence of betaine and É‘B-crystallin could provide mechanistic insight. 

I have some concerns about the data quality and reproducibility presented in this manuscript:

The aggregation curves in figure 1 are missing a negative protein control such as Lysozyme or BSA. This should be included.

In figure 3 the data form figure 2 is analyzed, but there are some inconsistencies and missing raw data. Figure 3 shows 7 data points, but only 5 curves are presented in figure 2, also the raw aggregation curve data for the aB-crystallin control (black dots) is missing. Furthermore, it appears that only one technical replicate is shown, the authors should provide at least 3 technical replicates to prove the reproducibility of the data and to allow us to make a robust assessment of the changes in the kinetic variables. Furthermore it would be good to provide the fits that were used to obtain the values in figure 3. This will allow the reader to better understand the changes in kinetic parameters.

The authors purify the É‘B-crystallin by size exclusion chromatography, but do not provide any information on the oligomeric state of É‘B-crystallin that was purified. A analytical ultracentrifugation experiment at room temperature could be informative to explore the difference between É‘B-crystallin at physiological and heat shock temperatures. Furthermore it would be good to know how betaine affects the oligomeric state of É‘B-crystallin at physiological temperature.

Minor comments:

In the first paragraph of the result section the authors say “...10 mM Arg in a mixture with a similar concentration of aB-Cr”, “Similar” it is not a scientific term that can be used for such an important parameter such as protein concentration please use more precise language.

The analytical ultracentrifugation data is missing information on the fits of the sedimentation velocity data. What was the frictional ratio fitted to the data? Did that change with different conditions ?

I think there is a mistake in the legend of figure 2 A. The olive curve says it is at 0.35 mg/ml but is less effective than the pink curve at 0.05 mg/ml.

In summary the paper provides an interesting insight into the interplay of chemical and molecular chaperones with the model protein Phb. The paper is lacking important experimental data. I would recommend publications after the above requested additional experimental data and technical replicates and controls are provided.

Author Response

Dear reviewer,

Thank you very much for careful reading of our paper and your valuable comments. Our reply is in the file.

Reviewer 3 Report

Overall, PPI's play an important role in a wide variety of cellular processes.  The interaction between a target protein and the chaperones is not widely explored and thanks to the authors for an in-depth investigation.

The methodology has been sound and results give an interesting insight into how crowding agents influence both the chaperones and the target protein. Considering the fact that crowding agents come closest to resembling the cellular environment, the MS makes for an interesting read.

The MS deserves to be accepted and rectifying some English statements aside, I do not see any trouble here.

Author Response

Dear reviewer,

Thank you very much for careful reading our paper.

Reviewer 4 Report

The submitted manuscript used several experimental tools to the effect of small-molecule chemical chaperones on the interaction with protein chaperones towards their target protein. It is an interesting but underrepresented topic in biochemistry and molecule biology. In detail, the authors showed that the effect of aB-crystallin on Phb protein’s aggregation can be regulated in a contrary way upon the presence of small molecule Betaine and Arginine. Moreover, Betaine stabilizes the aB-Cr while destabilizing the aB-Cr. The overall data is clearly presented, and the manuscript was well written. I only have several suggestions in the order of appearance.  

1). On page 2, in the third paragraph, the aB-crystallin was not fully typed in. Formats problem happen in other places as well, especially heavy in page 13.  It’s better if the authors go through the whole context one more time for phrasing and grammar etc.

2). For figures, I find it unnecessary and a bit confusing to use numbers to describe curves, in the condition that they are already marked with different colors (Figure 1, 5, 6, 8). In my experience, numbers are not preferred as a mark of plots simply to avoid the risk of confusion with other number-based quantities in the figure. For example, in figure 6/8, the numbers marked there can cause confuses. For these numbers, maybe consider to use italic and color, with different font sizes from both the axes.

3). In page 9, I suggest the addition of the sedimentation coefficient formula and add some contexts to better interpret results in Fig 6. and the AUC experiment.

4). Because of the rich data and results, I suggest the authors adding a short conclusion session for a more friendly need to readers.

5*). Lastly, key findings claimed are supported by kinetic experiments. I wonder how reproducible these experimental data are. If multiple measurements had been undertaken, is it possible to add error bars somewhere? I am not surging the authors to repeat experiments for this study but at least it’s better if you add related discussions.

Author Response

Dear reviewer,

We are thankful to you for the valuable comments and careful reading our article. We have taken into account all the comments and tried to answer them.

Round 2

Reviewer 1 Report

The authors have responded to my comments and placed the work in context in the supporting letter but they haven't made these changes to the manuscript. This extra detail should be added in.

Reviewer 2 Report

I thank the authors for responding to my review in such great detail. The authors have addressed my concerns.